# Distilling Reinforcement Learning Algorithms for In-Context Model-Based Planning

**Jaehyeon Son**[1]**, Soochan Lee**[2]**, Gunhee Kim**[1]
[1]Seoul National University, [2]LG AI Research
`sjh9876@snu.ac.kr, soochan.lee@lgresearch.ai, gunhee@snu.ac.kr`

## Abstract

Recent studies have shown that Transformers can perform in-context reinforcement learning (RL) by imitating existing RL algorithms, enabling sample-efficient adaptation to unseen tasks without parameter updates. However, these models also inherit the suboptimal behaviors of the RL algorithms they imitate. This issue primarily arises due to the gradual update rule employed by those algorithms. Model-based planning offers a promising solution to this limitation by allowing the models to simulate potential outcomes before taking action, providing an additional mechanism to deviate from the suboptimal behavior. Rather than learning a separate dynamics model, we propose Distillation for In-Context Planning (DICP), an in-context model-based RL framework where Transformers simultaneously learn environment dynamics and improve policy in-context. We evaluate DICP across a range of discrete and continuous environments, including Darkroom variants and Meta-World. Our results show that DICP achieves state-of-the-art performance while requiring significantly fewer environment interactions than baselines, which include both model-free counterparts and existing meta-RL methods. The code is available at https://github.com/jaehyeon-son/dicp.

## 1 Introduction

Since the introduction of Transformers (Vaswani et al., 2017), their versatility in handling diverse tasks has been widely recognized across various domains (Brown et al., 2020; Dosovitskiy et al., 2021; Bubeck et al., 2023). A key aspect of their success is in-context learning (Brown et al., 2020), which enables models to acquire knowledge rapidly without explicit parameter updates through gradient descent. Recently, this capability has been explored in reinforcement learning (RL) (Chen et al., 2021; Schulman et al., 2017; Lee et al., 2022; Reed et al., 2022), where acquiring skills in a sample-efficient manner is crucial. This line of research naturally extends to meta-RL, which focuses on leveraging prior knowledge to quickly adapt to novel tasks.

In this context, Laskin et al. (2023) introduce Algorithm Distillation (AD), an in-context RL approach where Transformers sequentially model the entire learning histories of a specific RL algorithm across various tasks. The goal is for the models to replicate the exploration-exploitation behaviors of the source RL algorithm, enabling them to tackle novel tasks purely in-context. Beyond replication, Laskin et al. (2023) show that this approach can enhance sample efficiency by bypassing intermediate learning steps or distilling histories from multiple actors. This combination of an off-the-shelf RL algorithm with the in-context learning capability of Transformers has demonstrated strong potential for improving the adaptability of meta-RL (Lee et al., 2023a; Liu & Abbeel, 2023; Huang et al., 2024; Zisman et al., 2024; Sinii et al., 2024).

However, prior in-context RL approaches have a notable limitation: they tend to replicate the suboptimal behaviors of the source algorithm. The source RL algorithm updates its policy gradually through gradient descent, sometimes deliberately preventing abrupt changes (Schulman et al., 2015; 2017). As a result, it may take multiple iterations to fully integrate newly discovered information, leading to repeated suboptimal actions during this process. Without a mechanism to deviate from the source algorithm's behavior, existing in-context RL methods (Laskin et al., 2023; Huang et al., 2024; Sinii et al., 2024) inherit these inefficiencies.

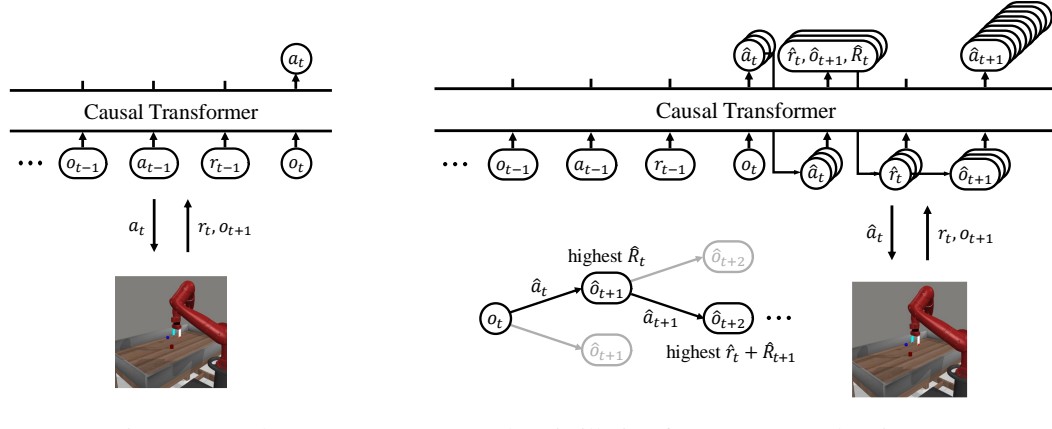

(a) Previous approaches        (b) Distillation for In-Context Planning (Ours)

Figure 1: Comparison between previous approaches (Laskin et al., 2023; Lee et al., 2023a; Huang et al., 2024) and our proposed Distillation for In-Context Planning (DICP). Unlike the previous approaches that directly predict actions without modeling dynamics, our approach leverages the in-context learned dynamics model for planning.

To address this limitation, we introduce *Distillation for In-Context Planning (DICP)*, an in-context model-based RL framework where the dynamics model is learned in-context alongside policy improvement. Unlike in-context policy improvement, in-context learning of the dynamics model does not inherit the suboptimal behaviors of the source algorithm, as modeling the environment's dynamics is independent of the inefficiency. By planning with this in-context learned dynamics model, our framework provides the agent with an additional mechanism to overcome the suboptimal behaviors of the source algorithm. Furthermore, simulating potential outcomes before executing actions allows for more deliberate decision-making based on predicted future returns. To the best of our knowledge, ours is the first model-based approach on in-context RL using Transformers to imitate the source algorithm.

To demonstrate the effectiveness of our framework, we conduct experiments in both discrete and continuous environments, including Darkroom variants (Laskin et al., 2023) and Meta-World benchmark suite (Yu et al., 2019). The results indicate that our approach significantly surpasses both in-context model-free counterparts (*e.g.*, AD (Laskin et al., 2023), DPT (Lee et al., 2023a), and IDT (Huang et al., 2024)) and existing meta-RL methods (*e.g.*, RL$^2$ (Duan et al., 2016), MAML (Finn et al., 2017), PEARL (Rakelly et al., 2019), MACAW (Mitchell et al., 2021), FOCAL (Li et al., 2021), BOReL (Dorfman et al., 2021), MoSS (Wang et al., 2023b), and IDAQ (Wang et al., 2023a)). Notably, our method achieves state-of-the-art results on the Meta-World ML1 and ML10 benchmarks, while requiring significantly fewer environmental interactions compared to the baselines.

## 2    RELATED WORK

**RL as sequence modeling.** With the advent of Transformers, which can learn from much larger datasets than what an agent can typically collect online, their application to offline RL (Levine et al., 2020) has gained prominence (Chen et al., 2021; Janner et al., 2021; Lee et al., 2022; Reed et al., 2022). Despite these advances, Laskin et al. (2023) point out a key limitation: these approaches struggle to improve their policy in-context through trial and error. The primary reason is that they are designed to imitate the dataset policy, which makes them unsuitable for performing in-context RL on novel tasks. To address this, Laskin et al. (2023) propose Algorithm Distillation (AD), an in-context RL approach where a Transformer is trained to distill learning histories from a source RL algorithm across diverse tasks. Notably, AD becomes effective when the context length of Transformers exceeds the episode horizon. Instead of imitating the source algorithm's actions, Decision-Pretrained Transformer (DPT; Lee et al. (2023a)) is designed to predict *optimal* actions. In-context Decision Transformer (IDT; Huang et al. (2024)) implements a hierarchical approach to in-context RL, where the Transformer predicts high-level decisions that guide a sequence of actions,

rather than predicting each action individually. In addition, Sinii et al. (2024) propose an in-context RL approach for variable action spaces, while Zisman et al. (2024) synthesize learning histories by gradually denoising policies rather than directly executing a source algorithm.

**Meta-RL.** Deep meta-RL began with online approaches, where RNNs are employed to learn RL algorithms that generalize across environments (Duan et al., 2016; Wang et al., 2016). In parallel, gradient-based approaches (Finn et al., 2017; Nichol et al., 2018) aim to discover parameter initializations that rapidly adapt to new tasks. Recently, *offline* meta-RL has gained attention, leveraging pre-collected meta-training datasets to address the meta-RL problem (Mitchell et al., 2021; Dorfman et al., 2021; Yuan & Lu, 2022; Wang et al., 2023a;b). Various methods have been explored for this problem, including gradient-based (Mitchell et al., 2021), Bayesian (Dorfman et al., 2021), and contrastive learning approaches (Yuan & Lu, 2022). Furthermore, recent in-context RL approaches (Laskin et al., 2023; Lee et al., 2023a; Huang et al., 2024; Sinii et al., 2024), including our own, fall within the category of offline meta-RL.

**Model-based meta-RL.** In the prior research on model-based meta-RL, Zintgraf et al. (2020; 2021); Dorfman et al. (2021) focus on learning to infer belief states about environment dynamics, while Wang et al. (2023b) aim to infer task representations. Wang et al. (2023a) quantify uncertainty using an ensemble of meta-training dynamics when facing novel meta-test tasks. Nagabandi et al. (2019); Pinon et al. (2022); Rimon et al. (2024) learn to construct dynamics models for planning. Our approach aligns with the latter category but stands out by integrating both the policy and the dynamics model within the same sequence model. This contrasts with prior work that either relies on separate modules for modeling dynamics (Nagabandi et al., 2019; Rimon et al., 2024), or omits a policy network while depending solely on the dynamics model (Pinon et al., 2022). We provide a more comprehensive overview of related works in App. C.

## 3 Problem Formulation

**POMDP.** We consider a partially observable Markov decision process (POMDP): $\mathcal{M} = (\mathcal{S}, \mathcal{A}, \mathcal{O}, \mathcal{T}, \mathcal{R}, \gamma)$. At each time step $t$, an agent interacts with the environment by selecting an action $a_t \in \mathcal{A}$ based on the current state $s_t \in \mathcal{S}$. The environment transitions to the next state $s_{t+1} \in \mathcal{S}$ according to the transition model $\mathcal{T}(s_{t+1} \mid s_t, a_t)$, and the agent receives a reward $r_t \in \mathcal{R}$, determined by the reward model $\mathcal{R}(r_t \mid s_t, a_t)$. However, the agent does not directly observe the true state $s_t$. Instead, it receives partial observation $o_t \in \mathcal{O}$ of the state $s_t$. The discount factor $\gamma \in [0, 1]$ controls the relative importance of immediate versus future rewards. The tuple $(\mathcal{T}, \mathcal{R})$ is referred to as the *dynamics model* or *world model*. The agent's *history* up to time step $t$ is defined as $h_t = (o_0, a_0, r_0, \ldots, o_t, a_t, r_t)$. The objective of RL is to find a policy that maximizes the expected cumulative reward $J = \mathbb{E}_{\pi, \mathcal{M}} \left[ \sum_{t=0}^{T} \gamma^t r_t \right]$, where $T$ is the number of interactions. Throughout the paper, we use the terms *environment* and *task* interchangeably to refer to a POMDP.

**Meta-RL.** We define an *algorithm* $f : \mathcal{H} \times \mathcal{O} \to \Delta(\mathcal{A})$, where $\mathcal{H}$ is the space of histories, and $\Delta(\mathcal{A})$ represents the space of probability distributions over $\mathcal{A}$. The objective of meta-RL is to discover an algorithm $f$ that maximizes the expected cumulative reward $J$ over a distribution of POMDPs $p(\mathcal{M})$. During *meta-training*, the algorithm is optimized on a set of tasks sampled from $p(\mathcal{M})$, while during *meta-test*, it is evaluated on another set of tasks sampled from $p(\mathcal{M})$. For simplicity, we refer to meta-training and meta-test as *training* and *test* when the context is clear.

Following previous works (Laskin et al., 2023; Huang et al., 2024), we focus on scenarios where an offline dataset of learning histories $\mathcal{D} = \{h_T^i = (o_0^i, a_0^i, r_0^i, \ldots, o_T^i, a_T^i, r_T^i)\}_{i=1}^{n}$ is available, generated by a source algorithm $f_{\text{source}}$ for a set of meta-training tasks $\{\mathcal{M}_i \sim p(\mathcal{M})\}_{i=1}^{n}$. The loss function for AD (Laskin et al., 2023) is defined as

$$\mathcal{L}_{\text{AD}}(\theta) = -\sum_{i=1}^{n} \sum_{t=1}^{T} \log f_\theta(a_t^i \mid o_t^i, h_{t-1}^i), \tag{1}$$

where $\theta$ represents a sequence model. Similarly, the loss functions for DPT (Lee et al., 2023a) and IDT (Huang et al., 2024) are respectively presented in Eq. 3-4 in App. C.

---

**Algorithm 1** Meta-Training Phase

---

1: **Required:** task distribution $p(\mathcal{M})$, source algorithm $f_{\text{source}}$, sequence model $\theta$, context length $k$
2: $\mathcal{D} \leftarrow \{\}$
3: **for** $i = 1 \ldots n$ **do**
4:     Sample a training task $\mathcal{M}_i \sim p(\mathcal{M})$.
5:     Execute $f_{\text{source}}$ on $\mathcal{M}_i$ and collect learning history $h_t^i = (o_0^i, a_0^i, r_0^i, \ldots, o_T^i, a_T^i, r_T^i)$.
6:     $\mathcal{D} \leftarrow \mathcal{D} \cup h_t^i$
7: **end for**
8: **while** not converge **do**
9:     Sample $k$-step segments from $\mathcal{D}$.
10:     Update $\theta$ for Eq. 2 with the sampled trajectory segments.
11: **end while**

---

**Algorithm 2** Meta-Test Phase

---

1: **Required:** task distribution $p(\mathcal{M})$, sequence model $\theta$
2: **for** $j = 1 \ldots m$ **do**
3:     Sample a test task $\mathcal{M}_j \sim p(\mathcal{M})$.
4:     $h_{-1} \leftarrow ()$
5:     **for** $t = 0 \ldots T$ **do**
6:         $a_t \leftarrow \text{DICP}(o_t, h_{t-1}, \theta)$
7:         $o_{t+1}, r_t \leftarrow$ Perform action $a_t$ on $\mathcal{M}_j$.
8:         $h_t \leftarrow (h_{t-1}, o_t, a_t, r_t)$
9:     **end for**
10: **end for**

---

**Algorithm 3** Distillation for In-Context Planning (DICP)

---

1: **Input:** Current observation $o_t$, learning history $h_{t-1}$, sequence model $\theta$
2: Set beam size $K$, sample size $L$
3: $\mathcal{B} \leftarrow \{\}$
4: **for** $l = 1 \ldots L$ **do**
5:     $\hat{a}_t \sim f_\theta(\,\cdot \mid o_t, h_{t-1})$
6:     $\hat{r}_t, \hat{o}_{t+1}, \hat{R}_t \sim g_\theta(\,\cdot \mid o_t, \hat{a}_t, h_{t-1})$
7:     $\hat{h}_t \leftarrow (h_{t-1}, o_t, \hat{a}_t, \hat{r}_t)$
8:     $\mathcal{B} \leftarrow \mathcal{B} \cup (\hat{h}_t, \hat{o}_{t+1})$
9: **end for**
10: **while** stopping criterion is not met **do**
11:     $\mathcal{B}' \leftarrow \{\}$
12:     **for all** $(\hat{h}_{s-1}, \hat{o}_s) \in \mathcal{B}$ **do**
13:         **for** $l = 1 \ldots L$ **do**
14:             $\hat{a}_s \sim f_\theta(\,\cdot \mid \hat{o}_s, \hat{h}_{s-1})$
15:             $\hat{r}_s, \hat{o}_{s+1}, \hat{R}_s \sim g_\theta(\,\cdot \mid \hat{o}_s, \hat{a}_s, \hat{h}_{s-1})$
16:             $\hat{h}_s \leftarrow (\hat{h}_{s-1}, \hat{o}_s, \hat{a}_s, \hat{r}_s)$
17:             $\mathcal{B}' \leftarrow \mathcal{B}' \cup (\hat{h}_s, \hat{o}_{s+1})$
18:         **end for**
19:     **end for**
20:     $\mathcal{B} \leftarrow$ Top $K$ elements of $\mathcal{B}'$ w.r.t. $\sum_{t'=t}^{s-1} \hat{r}_{t'} + \hat{R}_s$
21: **end while**
22: $(h_{t-1}, o_t, \hat{a}_t, \hat{r}_t, \ldots, \hat{o}_{S+1}) \leftarrow$ Top element of $\mathcal{B}$ w.r.t. $\sum_{t'=t}^{S-1} \hat{r}_{t'} + \hat{R}_S$
23: **Return:** $\hat{a}_t$

---

## 4   DISTILLATION FOR IN-CONTEXT PLANNING

In this section, we propose *Distillation for In-Context Planning (DICP)*, where the agent utilizes an in-context learned dynamics model to plan actions. To learn the dynamics model in-context, we introduce *meta-model* $g_\theta : \mathcal{H} \times \mathcal{O} \times \mathcal{A} \rightarrow \Delta(\mathbb{R}) \times \Delta(\mathcal{O}) \times \Delta(\mathbb{R})$, which shares the same sequence model $\theta$ with the algorithm $f_\theta$. It yields probability distributions over the spaces of reward $r_t$, next observation $o_{t+1}$, and return-to-go $R_t = \sum_{t'=t}^{H} r_{t'}$ based on history $h_{t-1}$, current observation $o_t$, and action $a_t$. $H$ represents the time step at which the episode ends. We optimize $\theta$ throughout the meta-training phase (Alg. 1) with following loss:

$$\mathcal{L}(\theta) = \mathcal{L}_{\text{Im}}(\theta) + \lambda \cdot \mathcal{L}_{\text{Dyn}}(\theta), \text{ where } \mathcal{L}_{\text{Dyn}}(\theta) = -\sum_{i=1}^{n} \sum_{t=1}^{T} \log g_\theta(r_t, o_{t+1}, R_t | o_t, a_t, h_{t-1}). \quad (2)$$

Here, $\mathcal{L}_{\text{Im}}$ corresponds to one of $\mathcal{L}_{\text{AD}}$, $\mathcal{L}_{\text{DPT}}$, or $\mathcal{L}_{\text{IDT}}$ (Eq. 1, 3-4), and $\lambda$ is a hyperparameter that balances the imitation loss $\mathcal{L}_{\text{Im}}$ and the dynamics loss $\mathcal{L}_{\text{Dyn}}$.

After meta-training, the sequence model $\theta$ is fixed and evaluated during the meta-test phase (Alg. 2). A key distinction between our method and previous in-context RL approaches (Laskin et al., 2023; Lee et al., 2023a; Huang et al., 2024) lies in how actions are selected (L6). Previous methods generate actions directly from the sequence model: $a_t \leftarrow f_\theta(\,\cdot \mid o_t, h_{t-1})$. These approaches may replicate the inefficiencies arising from the gradual updates of gradient-based RL algorithms, as they imitate the source algorithm's behaviors.

In contrast, our method employs an additional DICP subroutine (Alg. 3) for action selection. As a planning strategy, we adopt Model Predictive Control (MPC) (Nagabandi et al., 2018; 2019; Sæmundsson et al., 2018). At each time step, the agent simulates future outcomes, selects the path that maximizes predicted return, and executes the first action in that path. Specifically, upon receiving an observation $o_t$, the agent samples $L$ candidate actions $\{\hat{a}_t^1, \ldots, \hat{a}_t^L\}$, and predicts the consequences

of the actions using the dynamics model, which are the next observations $\{\hat{o}_{t+1}^1, \ldots, \hat{o}_{t+1}^L\}$ and rewards $\{\hat{r}_t^1, \ldots, \hat{r}_t^L\}$. The agent also predicts return-to-go $\{\hat{R}_t^1, \ldots, \hat{R}_t^L\}$ for each candidate action to evaluate each path. The process repeats for future time steps, generating new candidate actions $\{\{\hat{a}_{t+1}^{1,1}, \ldots, \hat{a}_{t+1}^{1,L}\}, \ldots, \{\hat{a}_{t+1}^{L,1}, \ldots, \hat{a}_{t+1}^{L,L}\}\}$ and corresponding consequences, until either the predefined planning horizon is reached or the episode ends. This process constructs a planning tree, where each path is ranked based on the predicted return. The overall DICP algorithm is presented in Alg. 3, where superscripts are omitted for brevity.

After building the planning tree, the agent can apply a suitable tree search algorithm that fits within computational and memory constraints. In our setup, we employ *beam search* as it is well-suited to Transformers, where decoding, sorting, and slicing operations for respective beams are parallelizable. For each planning step, the top $K$ paths are selected based on the predicted return. As demonstrated in §5, when the environment provides well-structured dense rewards, a simple *greedy search* combined with the in-context learned dynamics model achieves competitive performance, although beam search remains applicable in such environments. In Alg. 3, greedy search is represented by skipping the while loop of L10-21.

The DICP algorithm provides a mechanism for the agent to deviate from the source algorithm's suboptimal behaviors and immediately pursue reward-maximizing actions. Since modeling the environment's dynamics is independent of the source algorithm's gradual update rule, the agent can leverage the in-context learned dynamics model to enhance the decision-making process. When the agent identifies promising transitions that contain information for achieving higher returns, the dynamics model assigns higher expected returns to prospective state-action pairs that align with this information. When such pairs are sampled (L6 & L14), the dynamics model assigns them higher returns (L7 & L15), encouraging further exploration toward these promising areas (L20 & L22).

**Architecture.** We evaluate our framework using architectures from AD (Laskin et al., 2023), DPT (Lee et al., 2023a), and IDT (Huang et al., 2024), incorporating modifications to include the additional loss term $\mathcal{L}_{\text{Dyn}}$ as shown in Eq. 2. These modifications are illustrated in Fig. 4 in App. A. We refer to the resulting models as DICP-AD, DICP-DPT, and DICP-IDT. For DICP-AD and DICP-DPT, we append the action $a_t$ to the input sequence and train the model to predict the reward $r_t$, next observation $o_{t+1}$, and return-to-go $R_t$ conditioned on $a_t$. Additionally, following Lee et al. (2023a), we compress each $(o, a, r)$ tuple into a single token before passing it through the Transformer. For DICP-IDT, the model predicts the results of actions within its Decisions-to-Go module, which interacts with the environment through low-level actions. However, we find that the original architecture struggles to propagate reward information, as the Reviewing-Decisions module encodes only observations and actions. To address this, we modify the Making-Decisions module to also encode rewards. For completeness, we review the original contributions of Lee et al. (2023a); Huang et al. (2024) in App. C.

**Distribution choice.** For the distribution class of $f_\theta$, we use a categorical distribution for discrete action spaces and a Gaussian distribution with diagonal covariance for continuous ones. For $g_\theta$, we apply the same approach, using a categorical distribution for discrete state spaces and rewards, and a Gaussian for continuous ones. Thus, $\mathcal{L}_{\text{Im}}(\theta)$ and $\mathcal{L}_{\text{Dyn}}(\theta)$ correspond to either cross-entropy loss or Gaussian negative log-likelihood, depending on the environment.

## 5 EXPERIMENTS

### 5.1 ENVIRONMENTS

We evaluate our DICP framework across a diverse set of environments. For discrete environments, we use Darkroom, Dark Key-to-Door, and Darkroom-Permuted, which are well-established benchmarks for in-context RL studies (Laskin et al., 2023; Lee et al., 2023a; Huang et al., 2024). For continuous ones, we test on the Meta-World benchmark suite (Yu et al., 2019).

**Darkroom.** Darkroom is a 2D discrete environment where the agent should locate a goal with limited observations. The state space consists of a $9 \times 9$ grid, and the action space includes five actions: up, down, left, right, and stay. The agent earns a reward of 1 whenever reaching the goal,

with 0 reward otherwise. The agent's observation is restricted to its current position, which makes it challenging to infer the goal's location. The agent always starts at the center of the grid, and the tasks are distinguished by varying goal locations, resulting in 81 distinct tasks. The tasks are divided into disjoint training and test sets, with a 90:10 split. Each episode has a horizon of 20 steps.

**Dark Key-to-Door.** Dark Key-to-Door is a variation of Darkroom, where the agent should find a key before reaching the goal. The agent receives a reward of 1 upon finding the key and an additional reward of 1 upon reaching the goal. Unlike Darkroom, the agent only earns a reward from the goal once per episode, with a maximum reward of 2 per episode. This environment features $81 \times 81 = 6561$ distinct tasks, each defined by different key and goal locations. The train-test split ratio is 95:5. The episode horizon is extended to 50 steps.

**Darkroom-Permuted.** Darkroom-Permuted is another variant of Darkroom, where the action space is randomly permuted. In this environment, the agent starts in a fixed corner of the grid, with the goal located in the opposite corner. There are $5! = 120$ unique tasks, each corresponding to a different permutation of the action space. Since the agent is tested on novel permutations, it must explore the environment to figure out the effects of each action. The train-test split ratio is the same as in Darkroom. The episode horizon is 50 steps.

**Meta-World.** Meta-World (Yu et al., 2019) is a robotic manipulation benchmark suite designed for meta-RL and multi-task RL. The suite includes 50 continuous control tasks and offers three distinct meta-RL benchmarks: ML1, ML10, and ML45. We focus on ML1, which provides 50 pre-defined seeds for both training and test for each task. Each seed represents a different initialization of the object, goal, and the agent. Although the agent receives dense rewards tailored to each specific task, its performance is evaluated based on the average success rate across test seeds, which differs from the dense reward structure. The episode horizon is 500 steps. We also report results on ML10 in App. B, where the goal is to learn from 10 training tasks and generalize to 5 test tasks.

## 5.2 BASELINES

**In-context RL methods.** We compare our approach with recent in-context model-free methods, including AD (Laskin et al., 2023), DPT (Lee et al., 2023a), and IDT (Huang et al., 2024). We follow the evaluation protocols of previous works (Laskin et al., 2023; Huang et al., 2024). For the source algorithms, we use PPO (Schulman et al., 2017) to generate learning histories. We collect data using 100 actors, accumulating a total of 100K to 1M environment steps, depending on the environment. The context length is set to four and ten episode horizons for discrete and continuous environments, respectively. We evaluate DPT only in Darkroom variants because they assume the availability of optimal actions, which are not applicable in general environments.

**Other meta-RL methods.** For model-free meta-RL methods, we include RL$^2$ (Duan et al., 2016), MAML (Finn et al., 2017), PEARL (Rakelly et al., 2019), MACAW (Mitchell et al., 2021), and FOCAL (Li et al., 2021). For the model-based methods, we compare against MuZero (Schrittwieser et al., 2019), BOReL (Dorfman et al., 2021), MoSS (Wang et al., 2023b), and IDAQ (Wang et al., 2023a). To ensure a fair comparison, we report IDAQ results using SAC as the source algorithm, rather than expert policies. The original BOReL leverages an oracle reward function, so we report results without it, leading to failure on most tasks as discussed in Dorfman et al. (2021); Wang et al. (2023a). MuZero is not originally designed for meta-RL, but we include it due to its strong performance after pre-training on meta-training tasks followed by fine-tuning on meta-test tasks (Anand et al., 2022). Due to the challenge in reproducing meta-RL approaches, we present the results reported in the original papers, following prior works (Anand et al., 2022; Wang et al., 2023b).

## 5.3 RESULTS

We begin by comparing our approach with model-free counterparts on the Darkroom, Dark Key-to-Door, and Darkroom-Permuted environments. The first row of Fig. 2 displays the learning curves for these environments. Under our unified configurations and implementations, the best algorithm among AD, DPT, and IDT varies according to environments. Nonetheless, our model-based approach consistently outperforms the model-free counterparts. For planning, we use beam search

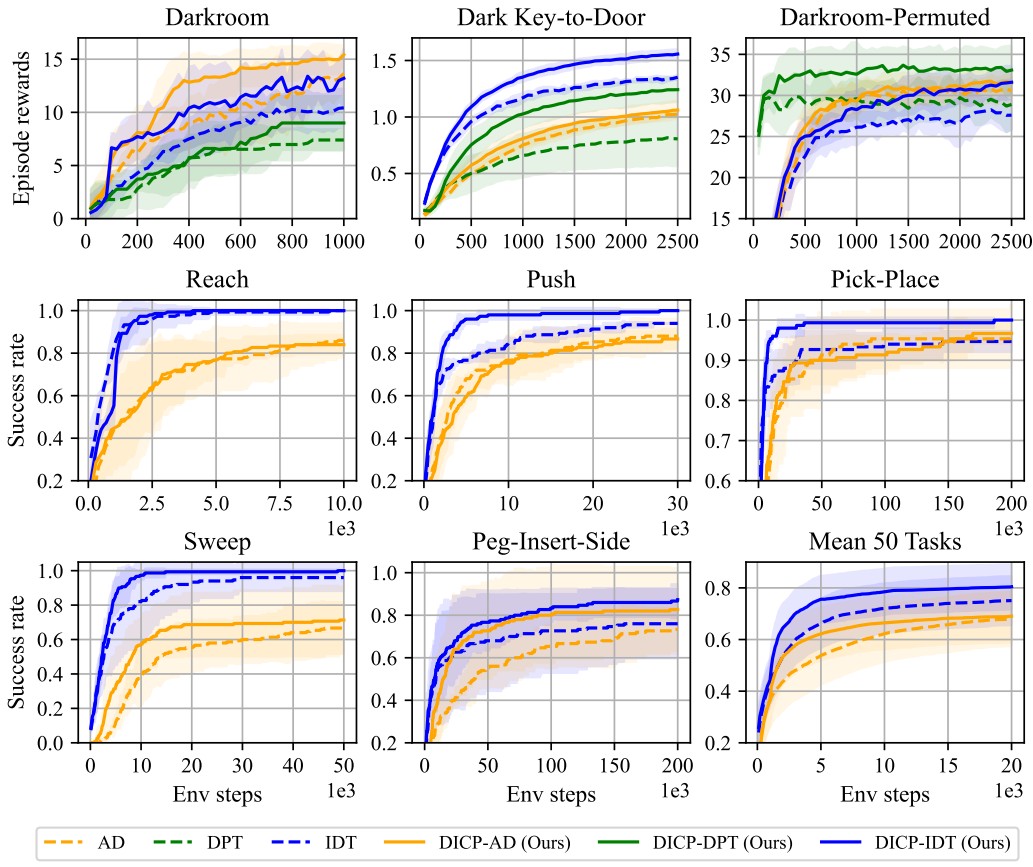

Figure 2: Learning curves of in-context RL approaches during the meta-test phase on discrete (1st row) and continuous (2nd and 3rd rows) environments. Our methods outperform model-free counterparts in both sample efficiency and overall performance. Results are averaged over 5 and 3 train-test splits for discrete and continuous benchmarks, respectively. We also report the mean success rate across all 50 tasks in Meta-World ML1. The final performance results for all ML1 benchmarks are presented in Table 1 and Table 8. Shaded areas represent 95% confidence intervals.

with a beam size of 10, considering all 5 actions at each planning step. The ablation study over different beam sizes are discussed in §6.1.

Furthermore, we evaluate our approach on Meta-World ML1 benchmarks against previous meta-RL methods, including both model-free and model-based methods. Table 1 presents the mean success rates over the meta-test phase for *Reach-v2*, *Push-v2*, *Pick-Place-v2*, *Sweep-v2*, and *Peg-Insert-Side-v2*, following previous works (Yu et al., 2019; Anand et al., 2022; Wang et al., 2023b;a). Our method achieves state-of-the-art performance on this benchmarks, surpassing all previous meta-RL methods. A key highlight is that our approach reaches this performance with a maximum of only 200K environment steps, whereas most other methods require at least 10M steps. To ensure a fair comparison with Wang et al. (2023a), whose meta-test is conducted with only 5K environment steps, we also report the results at 5K steps in Table 9. Our approach consistently outperforms the baselines from Wang et al. (2023a) at this step count as well. While in-context model-free baselines (Laskin et al., 2023; Huang et al., 2024) are also relatively sample-efficient, their performance remains suboptimal compared to our method. The learning curves are displayed in the second and third rows of Fig. 2. For planning, we employ greedy search with a sample size of 10 at each step. We also report additional results on all remaining ML1 benchmarks in Table 8 and on ML10 in Table 10, respectively.

Table 1: Meta-test success rates on Meta-World ML1. The success rates are reported as percentages, averaged over 3 train-test splits with 50 test seeds. The results marked with [*] are taken from Yu et al. (2019), while the ones with [†] are from Wang et al. (2023a). The results for MoSS and MuZero are from Wang et al. (2023b) and Anand et al. (2022), respectively. Our approach achieve state-of-the-art performance with much fewer environment steps. Results at 5K steps are presented in Table 9. We omit '-v2' from the task names.

| Method | Reach | Push | Pick-Place | Sweep | Peg-Insert-Side | Max Steps |
|---|---|---|---|---|---|---|
| RL[2][*] | **100** | 96 | 98 | – | – | 300M |
| MAML[*] | **100** | 94 | 80 | – | – | 300M |
| PEARL[*] | 68 | 44 | 28 | – | – | 300M |
| MACAW[†] | – | – | – | 4 | 0 | 5K |
| FOCAL[†] | – | – | – | 38 | 10 | 5K |
| MuZero | **100** | **100** | **100** | – | – | 10M |
| MoSS | 86 | **100** | **100** | – | – | 40M |
| BoREL[†] | – | – | – | 0 | 0 | 5K |
| IDAQ[†] | – | – | – | 59 | 30 | 5K |
| AD | 86 | 88 | 96 | 67 | 73 | 200K |
| IDT | **100** | 94 | 95 | 96 | 76 | 200K |
| DICP-AD (Ours) | 84 | 87 | 97 | 71 | 83 | 200K |
| DICP-IDT (Ours) | **100** | **100** | **100** | **100** | **87** | 200K |

## 6 ABLATION STUDY

We perform an ablation study to evaluate key design choices in our approach. We examine the effect of model-based planning at different scales (§6.1), the effect of context length on the accuracy of the in-context learned dynamics model (§6.2), and the effect of different source algorithms (§6.3).

### 6.1 EFFECT OF MODEL-BASED PLANNING AT DIFFERENT SCALES

We evaluate our approach using search algorithms at varying scales, as shown in the first row of Fig. 3. Experiments are conducted in two environments: Darkroom (discrete) and Reach-v2 (continuous). We use DICP-AD for Darkroom and DICP-IDT for Reach-v2, the best-performing models for each environment. The results demonstrate that incorporating model-based planning significantly improves performance compared to baselines that train the dynamics model but do not use it for planning. While increasing the beam size or sample size improves performance, the gains plateau after a size of 10. When the dynamics model is not used for planning, the performance is comparable to that of model-free baselines.

### 6.2 EFFECT OF CONTEXT LENGTHS ON ACCURACY OF THE DYNAMICS MODEL

We investigate the effect of context length on the accuracy of the in-context learned dynamics model. Our hypothesis is that extending the context length enables the model to capture more information about the environment, improving prediction accuracy. To verify this, we evaluate the test loss by forwarding the learning histories of the source algorithm on *meta-test* tasks into the meta-trained DICP-IDT on Reach-v2. The test loss is measured as the mean squared error between the predicted and actual rewards and observations. The models are trained with a context length of 1000, and the loss is calculated by averaging across every 10 positions within the context. The results in the second row of Fig. 3 support our hypothesis: longer context lengths yield more accurate predictions for both observations and rewards. Given that the effectiveness of model-based planning heavily depends on the dynamics model's bias (Janner et al., 2019; Hiraoka et al., 2021), our framework benefits from longer context lengths. Furthermore, as in-context model-free RL methods have also been shown to improve with extended context lengths (Laskin et al., 2023), our combined framework gains substantial advantages from using longer context lengths.

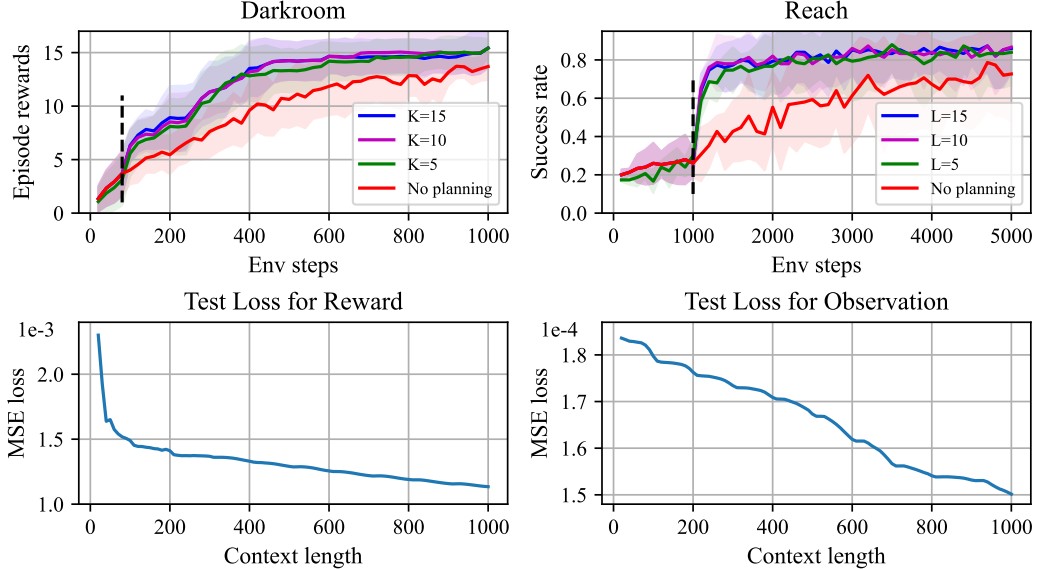

Figure 3: First row: The effect of model-based planning at different scales. We present learning curves with varying beam sizes $K$ and sample sizes $L$. For the case labeled "No planning," the dynamics model is not utilized for planning, while the meta-model is still trained. The dashed vertical line marks the time step when planning begins, coinciding with the point where the context is fully filled. Second row: The effect of context lengths on the accuracy of the in-context learned dynamics model. Results are averaged over 3 train-test splits.

Table 2: Meta-test success rates of in-context RL approaches using SAC as the source algorithm. Our approach outperforms original model-free IDT. The results are averaged over 3 train-test splits.

| Method | Reach | Push | Pick-Place | Sweep | Peg-Insert-Side |
|---|---|---|---|---|---|
| IDT | **85** | 65 | 17 | **4** | 1 |
| DICP-IDT (Ours) | **85** | **69** | **20** | 3 | **3** |

## 6.3 Effect of Different Source Algorithms

Our framework is agnostic to the choice of the source algorithm, as are previous model-free approaches (Laskin et al., 2023; Lee et al., 2023a; Huang et al., 2024). To empirically verify this, we train IDT and DICP-IDT with SAC (Haarnoja et al., 2018) as the source algorithm. As shown in Table 2, our method continues to outperform model-free counterpart. These results demonstrate that our approach is robust to the choice of source algorithm, outperforming model-free baselines regardless of which source algorithm is used. It is worth noting that since our SAC configuration is not specifically tuned for these tasks, the overall performance may be underestimated. The hyperparameters for SAC are detailed in App. A.

## 7 Conclusion

We introduced an in-context model-based RL framework that leverages Transformers to not only learn dynamics models but also improve policies in-context. By incorporating model-based planning, our approach effectively addresses the limitations of previous in-context model-free RL methods, which often replicate the suboptimal behavior of the source algorithms. Our framework demonstrated superior performance across various discrete and continuous environments, establishing in-context RL as one of the most effective meta-RL approaches.

One limitation of our approach is the additional computational cost incurred during action selection compared to model-free methods. However, this trade-off aligns with the broader trend of leveraging increased inference-time computation to maximize the reasoning capabilities of Transformers (Brown et al., 2020; Wei et al., 2022). Future work could explore adaptive planning strategies that dynamically adjust the planning scale based on context. Another promising direction is incorporating expert demonstrations to accelerate learning. Additionally, investigating offline dataset construction strategies to enable our method to adapt to changing dynamics would be a meaningful research direction (Lee et al., 2023b; 2024; Son et al., 2024). Finally, exploring more advanced or efficient sequence models could further enhance in-context RL.

## REPRODUCIBILITY STATEMENT

We are committed to ensuring the full reproducibility of our research. Our open-sourced code enables easy replication of all experiments, and all datasets used in our experiments can be generated using our code. Detailed configurations are provided in both the code and App. A. We hope this resource serves as a valuable reference for the research community, particularly for newcomers, by providing a unified implementation of in-context RL methods.

## ACKNOWLEDGMENTS

We thank Jaekyeom Kim, Yeda Song, Jinuk Kim, Fatemeh Pesaran zadeh, Wonkwang Lee, Dongwook Lee, and the anonymous reviewers for their thoughtful feedback. This work was partly supported by Samsung Advanced Institute of Technology, Institute of Information & Communications Technology Planning & Evaluation (IITP) grant funded by the Korea government (MSIT) (No. RS-2022-II220156, Fundamental research on continual meta-learning for quality enhancement of casual videos and their 3D metaverse transformation; No. RS-2019-II191082, SW StarLab; No. RS-2021-II211343, Artificial Intelligence Graduate School Program (Seoul National University)), and Korea Radio Promotion Association (Development of Intelligent Docent Service for Information-Disadvantaged Groups). Gunhee Kim is the corresponding author.

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

# A  IMPLEMENTATION DETAILS

## A.1  SOURCE ALGORITHMS

For the source algorithms (Schulman et al., 2017; Haarnoja et al., 2018), we use the implementation of Stable Baselines 3 (Raffin et al., 2021). The hyperparamter settings are detailed in Table 3-4. Unless otherwise specified, most hyperparameters are set to default values.

Table 3: Hyperparamters for the source PPO algorithm.

| Hyperparmater | Darkroom | Dark Key-to-Door | Darkroom-Permuted | Meta-World |
|---|---|---|---|---|
| learning_rate | 3e-4 | 3e-4 | 3e-4 | 3e-4 |
| n_steps | 20 | 50 | 50 | 100 |
| batch_size | 50 | 100 | 50 | 200 |
| n_epochs | 20 | 10 | 20 | 20 |
| $\gamma$ | 0.99 | 0.99 | 0.99 | 0.99 |
| total_timesteps | 100K | 100K | 100K | 1M |

Table 4: Hyperparamters for the source SAC algorithm.

| Hyperparmater | Meta-World |
|---|---|
| learning_rate | 3e-4 |
| learning_starts | 100 |
| batch_size | 128 |
| train_freq | 10 |
| gradient_steps | 1 |
| buffer_size | 1M |
| $\gamma$ | 0.99 |
| total_timesteps | 1M |

## A.2  TRANSFORMERS

We implement Transformers using the open-source TinyLlama (Zhang et al., 2024). The hyperparameters are provided in Table 5, along with additional parameters specific to IDT.

Table 5: Hyperparamters for Transformers. All three Transformer modules in IDT share the same set of hyperparameters.

| Hyperparmater | AD/DPT (disc.) | IDT (disc.) | AD (cont.) | IDT (cont.) |
|---|---|---|---|---|
| n_layer | 4 | 4 | 4 | 4 |
| n_head | 4 | 4 | 8 | 4 |
| n_embed | 32 | 32 | 64 | 32 |
| intermediate_size | 128 | 128 | 256 | 128 |
| dropout | 0.1 | 0.1 | 0.1 | 0.1 |
| attention_dropout | 0.1 | 0.1 | 0.1 | 0.1 |
| optimizer | AdamW | AdamW | AdamW | AdamW |
| scheduler | cosine decay | cosine decay | cosine decay | cosine decay |
| learning_rate (at start) | 1e-2 | 1e-3 | 1e-2 | 1e-3 |
| $\beta_1$ | 0.9 | 0.9 | 0.9 | 0.9 |
| $\beta_2$ | 0.99 | 0.99 | 0.99 | 0.99 |
| weight_decay | 0.01 | 0.01 | 0.01 | 0.01 |
| $\lambda$ | 1 | 1 | 1 | 1 |
| n_actions per high-level decision | – | 10 | – | 10 |
| dim_z | – | 8 | – | 8 |

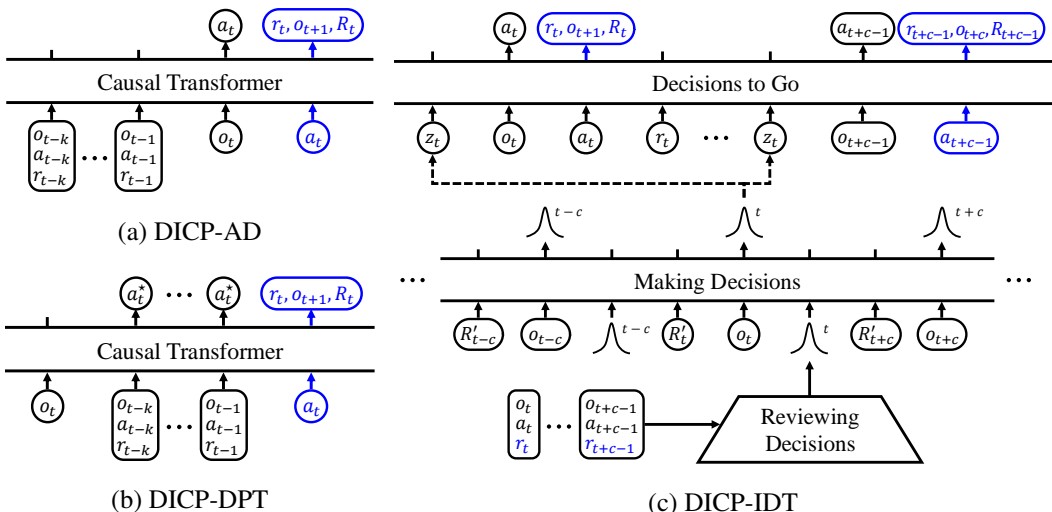

Figure 4: Model-based adaptation of in-context RL methods for DICP. The notations are defined as follows: $z$: high-level decisions, $R$: return-to-go, $k$: the number of transitions within the context, $c$: the number of low-level actions guided by a single high-level decision, and $R'$: relabeled return-to-go, following (Huang et al., 2024). Newly introduced components are highlighted in blue.

Table 6: Hyperparamters for planning.

| Hyperparmater | Darkroom | Dark Key-to-Door | Darkroom-Permuted | Meta-World |
|---|---|---|---|---|
| planning horizon | 8 | 16 | 16 | 1 |
| beam size ($K$) | 10 | 10 | 10 | – |
| sample size ($L$) | 5 | 5 | 5 | 10 |

## A.3 ENVIRONMENTS

In the ML1 benchmarks, we observe that using shorter horizons for the source algorithm accelerates the learning of in-context RL methods. Specifically, we terminate the source algorithm after 100 steps, as opposed to the original 500-step horizon provided by Meta-World. To minimize train-test discrepancies, we also meta-test the in-context RL methods using 100-step horizons. Even under these less favorable conditions, in-context RL methods still outperform the baselines, as shown in Fig. 2 and Table 1.

## A.4 PLANNING

We find that predicting only the immediate reward, rather than both the immediate reward and the return-to-go, is sufficient to outperform baselines. Summation of predicted immediate rewards up to the current step can serve as a myopic proxy for the total return of a planning path. In Darkroom variants, the binary nature of rewards often leads to different beam paths achieving the same

Table 7: FLOPs per action selection.

| Method | Darkroom | Dark Key-to-Door | Darkroom-Permuted | Meta-World |
|---|---|---|---|---|
| AD | 6M | 20M | 20M | 709M |
| DPT | 6M | 20M | 20M | 709M |
| IDT | 8M | 8M | 8M | 3M |
| DICP-AD | 2G | 18G | 18G | 8G |
| DICP-DPT | 2G | 18G | 18G | 8G |
| DICP-IDT | 147M | 147M | 147M | 15M |

cumulative reward during planning. When such ties arise, we break them by following the model's preference during action generation. This approach allows the agent to behave like a model-free counterpart when the goal or key is distant, while benefiting from model-based planning when the dynamics model can confidently predict rewards over shorter horizons. In Meta-World, the dense, human-designed reward structure further improves the effectiveness of this myopic return estimation. Since the rewards are designed to guide the agent toward the goal by greedily following reward increases, ranking planning paths based on immediate rewards often sufficient to improve in-context RL approaches. Hyperparamters for planning are detailed in Table 6.

Table 8: Results on all benchmarks of Meta-World ML1. We omit the rows for the 5 tasks reported in Table 1, while they are still included in the mean calculation.

| Task | AD | IDT | DICP-AD (Ours) | DICP-IDT (Ours) |
|---|---|---|---|---|
| Assembly | 0 | 2 | **28** | 4 |
| Basketball | 50 | 54 | 48 | **66** |
| Bin-Picking | 2 | **6** | 0 | **6** |
| Box-Close | 34 | 32 | **44** | 38 |
| Button-Press-Topdown | 98 | **100** | **100** | **100** |
| Button-Press-Topdown-Wall | 98 | **100** | **100** | **100** |
| Button-Press | **100** | **100** | **100** | **100** |
| Button-Press-Wall | **100** | **100** | **100** | **100** |
| Coffee-Button | **100** | **100** | **100** | **100** |
| Coffee-Pull | 88 | 78 | 42 | **90** |
| Coffee-Push | 98 | **100** | 92 | **100** |
| Dial-Turn | 76 | 76 | 30 | **94** |
| Disassemble | 0 | 0 | 0 | 0 |
| Door-Close | **100** | **100** | **100** | **100** |
| Door-Lock | 98 | **100** | 88 | 98 |
| Door-Open | 78 | 14 | **90** | 80 |
| Door-Unlock | 78 | 72 | 74 | **82** |
| Hand-Insert | 84 | 70 | **86** | 68 |
| Drawer-Close | **100** | **100** | **100** | **100** |
| Drawer-Open | 26 | 46 | 62 | **90** |
| Faucet-Open | **100** | **100** | 96 | 98 |
| Faucet-Close | **100** | **100** | 86 | **100** |
| Hammer | 8 | 22 | **52** | 10 |
| Handle-Press-Side | **100** | **100** | 94 | **100** |
| Handle-Press | **100** | **100** | **100** | **100** |
| Handle-Pull-Side | 36 | **100** | 82 | **100** |
| Handle-Pull | 56 | 66 | 74 | **78** |
| Lever-Pull | 96 | 96 | **100** | 98 |
| Pick-Place-Wall | 6 | 66 | 6 | **70** |
| Pick-Out-Of-Hole | 36 | **98** | 2 | 86 |
| Plate-Slide | 92 | **96** | 94 | 90 |
| Plate-Slide-Side | **100** | **100** | **100** | **100** |
| Plate-Slide-Back | 92 | **98** | 88 | 86 |
| Plate-Slide-Back-Side | 94 | **100** | **100** | 96 |
| Peg-Unplug-Side | 86 | **100** | 38 | 98 |
| Soccer | 94 | 96 | 92 | **100** |
| Stick-Push | **2** | 0 | 0 | **2** |
| Stick-Pull | 2 | 14 | 0 | **50** |
| Push-Wall | 86 | 94 | 92 | **100** |
| Push-Back | 0 | 0 | 0 | 0 |
| Reach-Wall | **100** | 98 | 94 | **100** |
| Shelf-Place | 0 | **68** | 0 | 62 |
| Sweep-Into | 86 | 86 | **88** | **88** |
| Window-Open | **100** | **100** | **100** | **100** |
| Window-Close | **100** | **100** | 88 | 98 |
| Mean | 68 | 75 | 69 | **80** |

While model-based planning requires additional computation, the cost is negligible. Specifically, our method does not increase the number of training parameters compared to model-free counter-

parts, and the primary difference lies in the increased number of Transformer inferences per action selection. As shown in Table 7, the maximum computation per action selection is approximately 18 GFLOPs in our experiment. Given that modern GPUs can process hundreds of TFLOPs per second, the computational expense is minimal in practice while the performance gains are substantial, making the trade-off highly favorable in our framework. The difference becomes even less significant when using architectures like IDT, which are specifically designed to handle longer sequences efficiently.

# B  ADDITIONAL EXPERIMENTS

We conducted experiments across all 50 environments of Meta-World ML1, using PPO (Schulman et al., 2017) as the source algorithm with the same hyperparameters as those used in Table 1. Each method is evaluated with 20K environment steps during the meta-test phase. As shown in Table 8, our method outperforms model-free counterparts. The mean learning curve is displayed in Fig. 2. It is noteworthy that since our configuration for the source algorithm is not tuned for every task, the results may not fully reflect the full potential of in-context RL methods, including ours.

Table 9: Meta-test success rates with 5K environment steps on 5 benchmarks of ML1. The results marked with [†] are taken from Wang et al. (2023a). Our approach consistently outperforms all the baselines at 5K steps as well.

| Method | Sweep | Peg-Insert-Side | Steps |
|---|---|---|---|
| MACAW[†] | 4 | 0 | 5K |
| FOCAL[†] | 38 | 10 | 5K |
| BoREL[†] | 0 | 0 | 5K |
| IDAQ[†] | 59 | 30 | 5K |
| IDT | 71 | 41 | 5K |
| DICP-IDT (Ours) | **87** | **45** | 5K |

Furthermore, we also conducted experiments on ML10 from Meta-World, a benchmark designed to evaluate the generalization ability of meta-RL methods using ten predefined training tasks and five test tasks. The results, presented in Table 10, demonstrate that our approach surpasses the model-free counterpart and achieves state-of-the-art performance on this benchmark with significantly fewer environment steps than the baselines. Notably, our method does not rely on expert demonstrations or task descriptions for the test tasks.

Table 10: Meta-test success rates on Meta-World ML10. The success rates are reported as percentages, averaged over 3 different seeds. The results marked with [*] are taken from Yu et al. (2019).

| Method | Success Rate | Steps |
|---|---|---|
| PEARL[*] | 13.0 | 350M |
| MAML[*] | 31.6 | 350M |
| RL$^2$[*] | 35.8 | 350M |
| IDT | 36.7 | 500K |
| DICP-IDT (Ours) | **46.9** | 500K |

# C  EXTENDED RELATED WORK

**DPT.**  Decision-Pretrained Transformer (DPT; Lee et al. (2023a)) is an in-context RL approach that meta-trains (or pre-trains) Transformers to predict *optimal* actions based on contextual trajectories. Consequently, DPT requires access to the optimal actions during training. The authors suggest several sources for gathering the meta-training dataset, including (i) randomly sampled trajectories directly from the environment, (ii) trajectories generated by a source RL algorithm, and (iii) trajectories of an expert policy. In this work, we concentrate on the second source to minimize reliance on

direct access to environment dynamics and expert policies. While the authors also discuss deploying DPT when offline datasets of the *meta-test* tasks are available, our focus is on scenarios where only offline datasets for *meta-train* tasks are available for a fair comparison. In this scheme, the loss function of DPT is defined as

$$\mathcal{L}_{\text{DPT}}(\theta) = -\sum_{i=1}^{n} \sum_{t=1}^{T} \log f_\theta(a_t^\star \mid o_t^i, h_{t-1}^i), \tag{3}$$

where $a_t^\star$ denotes the optimal action on the true state $s_t$.

**IDT.** In-Context Decision Transformer (IDT; Huang et al. (2024)) addresses the high computational costs of previous in-context RL methods, particularly when handling long-horizon tasks. IDT restructures the decision-making process to predict high-level decisions instead of individual actions. These high-level decisions then guide multiple action steps in a separate module, dividing the Transformer used in the earlier approach (Laskin et al., 2023) into modules with shorter sequences. The architecture of IDT consists of three modules: (i) the Making-Decisions module predicts high-level decisions, (ii) the Decisions-to-Go module decodes these high-level decisions into low-level actions, and (iii) the Reviewing-Decisions module encodes the resulting low-level actions back into high-level decisions. The loss function for IDT is structured similarly to that of AD, while the loss can be divided across the three modules:

$$\begin{aligned}
\mathcal{L}_{\text{IDT}}(\theta) &= -\sum_{i=1}^{n} \sum_{t=1}^{T} \log f_\theta(a_t^i \mid o_t^i, h_{t-1}^i) \\
&= -\sum_{i=1}^{n} \sum_{t=1}^{T} \left[ \log f_\theta(a_t^i \mid z_t, o_t^i) + \log f_\theta(z_t \mid o_t^i, \hat{z}_{t-1}) + \log f_\theta(\hat{z}_{t-1} \mid h_{t-1}^i) \right], \tag{4}
\end{aligned}$$

where $z_t$ is a high-level decision and $\hat{z}_{t-1}$ is an encoded context by the Reviewing-Decisions module.

**TTO.** TTO (Janner et al., 2021) is an offline RL approach that imitates the policy of the offline dataset, while simultaneously learning a dynamics model to facilitate planning to improve upon that policy. Our approach is similar to this in that we also use model-based planning to improve upon offline datasets. However, our framework focuses on improving a distilled algorithm rather than a specific policy. This distinction makes our problem setting more general and challenging compared to single-task RL like TTO, since our agent should learn the dynamics model primarily in-context when encountering novel tasks.

**Model-based RL.** Model-based RL has gained significant attention due to its ability to improve sample efficiency by utilizing a dynamics model to generate simulated experiences (Ha & Schmidhuber, 2018; Kaiser et al., 2020; Janner et al., 2019; Schrittwieser et al., 2019; Ye et al., 2021; Hafner et al., 2023). Some approaches further incorporate planning with search algorithms, such as MCTS (Schrittwieser et al., 2019; 2021; Ye et al., 2021) or beam search (Janner et al., 2021), to explore the planning space effectively. However, a key challenge in model-based RL is managing the balance between sample efficiency and model bias, as errors in the learned dynamics model can lead to suboptimal decision-making (Janner et al., 2019).

**Model-based Meta-RL.** Previous approaches have explored model-based approach to meta-RL (Sæmundsson et al., 2018; Lin et al., 2020; Hiraoka et al., 2021; Rimon et al., 2022). Clavera et al. (2018) aim to overcome the limitations of model-based RL (Janner et al., 2019) by incorporating a meta-model. Nagabandi et al. (2019) improve upon model-free meta-RL methods (Duan et al., 2016; Finn et al., 2017) to adapt shifting dynamics. Anand et al. (2022) focus on zero-shot generalization, adapting MuZero (Schrittwieser et al., 2019) to meta-RL. More recently, Rimon et al. (2024) introduce a model-based meta-RL approach that combines a meta-RL method (Zintgraf et al., 2020) with a model-based RL method (Hafner et al., 2023).

