# OpenReview forum: "Distilling Reinforcement Learning Algorithms for In-Context Model-Based Planning"
_ICLR.cc/2025/Conference — ICLR 2025 Poster_

### Official Review · Reviewer_zBei · 2024-10-27

**Soundness:** 4
**Presentation:** 4
**Contribution:** 4
**Rating:** 8
**Confidence:** 3

**Summary:**

This paper extends the use of decision transformers for in-context meta-task learning to incorporate model-based planning. The main innovation here is to have the transformer output predicted state values (r, o, R) in addition to the next action, and to use this state-transition model to select better actions. This can be applied to multiple different transformer-based agents and yields improvements both in terms of sample efficiency and overall score.

**Strengths:**

- Advances the performance of in-context RL, an exciting recent direction
- Provides a mechanism to overcome suboptimal behaviors inherited from the source RL algorithms
- Achieves state of the art on Meta-World, compared with a large variety of RL and in-context RL algorithms
- This is an important new innovation that makes sense, and as far as I can tell (though I do not have full knowledge of the literature) is the first demonstration of incorporating model-based planning into transformer-based in-context RL.
- This seems like a well-done paper with a straightforward but impactful contribution.

**Weaknesses:**

- Nothing major.

**Questions:**

- How exactly are the actions sampled, and what is the sensitivity to the sampling approach?

- What do you think would happen if the imitation loss and dynamics loss trained two separate models?

- I would be interested in more discussion of how this method might apply to online learning. For example, how might it interact with intrinsically-rewarded exploration to improve the world model? How much does this method depend on the quality of the offline dataset? How effectively would this approach adapt to an environment where the dynamics change?

- Would MCTS perhaps have benefits relative to beam search? Is there a way to not build the whole planning tree as an initial step, or what is the advantage to doing so?

- Are there scenarios where having a world model might detract. For example, what happens if the world model is not accurate enough?

- What are possible explanations for why model-based performs worse on Pick-Out-Of-Hole?

---

> ### Author Response · Authors · 2024-11-19
> **Author Response to Reviewer zBei**
>
> We greatly appreciate Reviewer zBei’s positive feedback and hope that this discussion contributes to advancing further research endeavors.
> ***
>
> ### Action Selection Mechanism and Sensitivity
>
> > Q. How exactly are the actions sampled, and what is the sensitivity to the sampling approach?
>
> The action selection process in our approach is as follows: the Transformer first predicts a distribution over actions, conditioned on the sequence of past transitions and the current observation. Multiple action candidates are sampled from this distribution. Each candidate action is appended to a duplicated input sequence, and these sequences are processed in parallel by the Transformer to predict the corresponding next observation, reward, and return-to-go. These predictions are further appended to the duplicated input sequences to sample candidate actions for subsequent steps. This process is repeated iteratively until a predefined planning horizon is reached. At the end of the process, the best action candidate from the first step is selected and executed, and the entire planning process is repeated. A detailed description of this algorithm, including the planning tree pruning method, can be found in Alg. 2-3 and Sec. 4.
>
> Regarding the choice of the sampling distribution, we experimented with several options, such as Gaussian distributions with unit and diagonal covariance matrices. We observed minimal performance differences between these configurations and opted for the diagonal covariance matrix for applicability. While alternative approaches, such as discretizing and sequentially predicting dimensions of the continuous action space (as in prior works), are feasible, we did not pursue them in this work due to their increased sequence length.
> ***
>
> ### Optimizing Imitation and Dynamics Losses with Separate Models
>
> > Q. What do you think would happen if the imitation loss and dynamics loss trained two separate models?
>
> As long as a single model has sufficient representational capacity, we do not anticipate any significant performance difference between using a single model versus separate models for imitation and dynamics losses. We consider a single sequence model to be a more practical choice, offering greater simplicity and flexibility, particularly when scaling, modifying, or deploying the model.
> ***
>
> ### Further Development of DICP
>
> > Q. I would be interested in more discussion of how this method might apply to online learning. For example, how might it interact with intrinsically-rewarded exploration to improve the world model?
>
> We agree that incorporating intrinsic rewards into our approach presents an exciting future direction. Since reward models are learned purely in-context within our framework, we anticipate that diverse transitions driven by intrinsic rewards could significantly enhance the accuracy of world model learning. Additionally, strategies inspired by language model decoding, such as repetition penalties, could potentially be adapted to seamlessly integrate intrinsic rewards into our method, making this an especially promising avenue for exploration.
>
> > Q. How much does this method depend on the quality of the offline dataset?
>
> The quality of the offline meta-training dataset is indeed critical, as our framework relies on in-context learning to train both the policy and the world model. Ensuring that the offline dataset captures well-structured and meaningful learning histories from the source algorithm is essential for the success of our method. This is analogous to the field of large language models, where improvements in dataset quality often lead to substantial performance gains.
>
> > Q. How effectively would this approach adapt to an environment where the dynamics change?
>
> Adapting our method to environments with changing dynamics is another exciting direction for future work. Building on our response to the previous question, we believe that collecting learning histories of source algorithms in environments with **changing dynamics** is essential for enabling effective adaptation at test time. If the Transformer is properly meta-trained on such an offline dataset, it is likely to perform robustly even in the face of dynamic changes.
> ***

---

> ### Author Response · Authors · 2024-11-19
> **Author Response to Reviewer zBei**
>
> ### MCTS and Planning Tree
>
> > Q. Would MCTS perhaps have benefits relative to beam search?
>
> MCTS is a sophisticated planning algorithm that is well-established in the model-based RL literature. By iteratively simulating outcomes from various nodes and making decisions based on aggregated results, MCTS tends to perform well in long-horizon planning and is particularly effective in stochastic dynamics. Additionally, its adaptive node expansion, rather than relying on a fixed number of leaf nodes, makes it especially suitable for larger action spaces.
>
> However, we opted for beam search in our work because its structure aligns well with the Transformer architecture. Beam search enables parallelized decoding, sorting, and slicing operations across beams, enhancing computational efficiency in such settings. Moreover, we can effectively manage GPU memory usage by adjusting the beam size, which provides practical benefits when deploying our method. Ultimately, the choice of planning algorithm depends on the environment and the computational budget.
>
> > Q. Is there a way to not build the whole planning tree as an initial step, or what is the advantage to doing so?
>
> In our approach, we avoid constructing the entire planning tree at the outset by pruning planning paths at every planning step. Specifically, planning paths are ranked using the predicted return as a value function, which is estimated by the Transformer. This value function integrates seamlessly with MCTS or other tree search algorithms, allowing us to circumvent the need to build an exponentially growing planning tree while maintaining computational efficiency and performance.
> ***
>
> ### Inaccuracy of World Model
>
> > Q. Are there scenarios where having a world model might detract? For example, what happens if the world model is not accurate enough?
>
> If the environment dynamics shift, the learned world model may not plan effectively, potentially leading to a suboptimal guidance for the agent. To address this issue, it is a promising direction to develop a mechanism for evaluating the accuracy of the world model at each step to adaptively decide when it should be relied upon. Another potential approach could involve constructing an offline meta-training dataset containing successful learning histories in the scenarios where inaccuracies are likely to occur.
> ***
> ### Inferior Performance in a Benchmark
>
> > Q. What are possible explanations for why model-based performs worse on Pick-Out-Of-Hole?
>
> Model-based planning generally provides a performance advantage when the world model is sufficiently accurate. The observed suboptimality likely arises from inaccuracies in the learned world model. Successful in-context learning of test dynamics depends on some degree of transferability between the training and test splits, which can vary across tasks. For instance, in tasks like Pick-Out-Of-Hole, the 50 training seeds may lack sufficient diversity to enable effective generalization of world model learning during the test. In such cases, one potential solution is to disable planning and rely on model-free action selection, similar to model-free counterparts. While this approach could mitigate the issue, we chose to omit it in our experiments to maintain consistency with the scope of our work.

---

> > ### Comment · Reviewer_zBei · 2024-11-22
> >
> > Thank you, I appreciate the answers to the questions. I remain supportive of this work being accepted.

---

> > > ### Author Response · Authors · 2024-11-23
> > > **Author Response to Reviewer zBei**
> > >
> > > We sincerely thank the reviewer for thoughtful engagement and continued support of our work. We hope this discussion translates into meaningful contributions within the community.

---

### Official Review · Reviewer_XR4p · 2024-10-29

**Soundness:** 3
**Presentation:** 3
**Contribution:** 3
**Rating:** 6
**Confidence:** 2

**Summary:**

The paper proposes a novel method called Distillation for In-Context Model-Based Planning (DICP) to improve the efficiency and effectiveness of in-context reinforcement learning. DICP leverages a learned dynamics model to predict the consequences of actions and uses this information to plan more effectively.

**Strengths:**

-Transformer simultaneously learns environment dynamics and improves policy in-context

-Avoid sub-optimal behavior of the source algorithm

**Weaknesses:**

-One of the flaws in model based planning is that the model might not be perfect. Errors of the world model might lead to sub-optimal policies as well which haven’t been discussed in the paper.

-Some analysis on how many times sub-optimal behvior from source algorithm was discovered and your approach was able to learn the optimal policy would be important to ensure extra computation of model based planning is worth it.

**Questions:**

1) Can you comment on how much the trade-off between performance vs computation is required in your approach and other comparisons? Do the gains outweigh computational expense?

2) What happens when the world model is incorrect? How is the performance affected? What steps are taken to ensure model-based planning can be accurate?

---

> ### Author Response · Authors · 2024-11-19
> **Author Response to Reviewer XR4p**
>
> We deeply appreciate reviewer XR4p's constructive feedback. We hope that our response addresses all of the reviewer's concerns clearly and comprehensively.
> ***
>
> ### Effect of Incorrect World Models
>
> As the reviewer points out, inaccuracy in the world model has been a significant concern in the model-based RL literature. Indeed, this inaccuracy could diminish the effectiveness of model-based approaches, including ours.
>
> We would like to clarify that we discussed the effect of world model inaccuracy and some ways to mitigate them in Sec. 6.2, where we conducted an ablation study on the relation between context lengths and world model accuracy, noting: “Given that the effectiveness of model-based planning heavily depends on the dynamics model’s bias [1, 2], our framework benefits from longer context lengths.” The ablation results show that longer context lengths, combined with sequence models with sufficient representational power, can be a great recipe for improving world model accuracy, which in turn enhances performance.
>
> Moreover, even in the scenarios where the in-context learned world models are not sufficiently accurate, our approach can maintain competitive performance by adopting the same action selection mechanism as model-free counterparts, without relying on the learned world models, as we empirically demonstrated in Sec. 6.1. As a meaningful future direction, model-based planning could be further improved by adaptively leveraging the world models based on their quantified accuracy at each decision-making step, which could also alleviate the reviewer’s concern.
>
> [1] Michael Janner, Justin Fu, Marvin Zhang, and Sergey Levine. When to trust your model: Model-based policy optimization. In Neural Information Processing Systems, 2019.
>
> [2] Takuya Hiraoka, Takahisa Imagawa, Voot Tangkaratt, Takayuki Osa, Takashi Onishi, and Yoshi-masa Tsuruoka. Meta-model-based meta-policy optimization. In Asian Conference on Machine Learning, 2021.
> ***
>
> ### Quantification of Sub-optimality
>
> As the reviewer suggested, proper quantification and comparative analysis are indeed crucial. We would like to clarify that in our framework, an “algorithm” is a meta-level concept that trains a policy rather than being a policy itself. As such, measuring sub-optimality in terms of “the number of times sub-optimal behavior is observed” may not be directly applicable to our approach. Instead, we believe the improvement of sub-optimality is more effectively quantified by assessing “the steepness of the learning curve,” which reflects the efficiency and capability of algorithms in training policies. As shown in Fig. 2, our approach achieves faster policy training with much fewer environment interactions compared to baseline methods in most cases. This demonstrates our method's ability to reduce sub-optimality effectively.
> ***
>
> ### Trade-off between Performance and Computation
>
> The trade-off between performance and computation is important to evaluate the practicality of performance improvement. We would like to emphasize that the additional computational cost in our framework is negligible. Specifically, our method does not increase the number of training parameters compared to model-free counterparts, and the primary difference lies in the increased number of Transformer inferences per action selection.
>
> In our experiments, the maximum computation per action selection is approximately 18 GFLOPs. Given that modern GPUs can process hundreds of teraFLOPs per second, this cost allows for action selection to occur thousands of times per second. Consequently, the computational expense is minimal in practice while the performance gains are substantial, making the trade-off highly favorable in our framework. We will add the related description in our paper to address this point.

---

> ### Comment · Reviewer_XR4p · 2024-11-20
>
> * The authors mention the effect of inaccurate world models but couldn't notice it in the results. Do you have any results where DICP didn't perform well due to this? Are there any other limitations?
>
> * If the difference between [1] and your approach lies in how actions are selected and  if your approach uses the DICP subroutine then it may be efficient, when mimicking the source algorithm is inefficient. But if the world model is inaccurate and the mimicking source algorithm is efficient then would it make DICP inefficient?
>     * In extension to this question: How often do inefficiencies of source algorithms cause inefficient learning in [1]?
>     * With enough data for in-context learning, would this problem persist?
>     * What may be other alternatives to a model-based planning approach to solve this problem and why would model-based planning be a better solution to this problem?
>
> * How does the computation of your approach compare to [1]?
>
> [1] Laskin M, Wang L, Oh J, Parisotto E, Spencer S, Steigerwald R, Strouse DJ, Hansen S, Filos A, Brooks E, Gazeau M. In-context reinforcement learning with algorithm distillation. arXiv preprint arXiv:2210.14215. 2022 Oct 25.

---

> ### Author Response · Authors · 2024-11-22
> **Author Response to Reviewer XR4p**
>
> We sincerely appreciate reviewer XR4p’s additional feedback.
> ***
>
> ### Effect of Incorrect World Models
>
> > Q. The authors mention the effect of inaccurate world models but couldn't notice it in the results. Do you have any results where DICP didn't perform well due to this?
>
> To address the reviewer’s inquiry regarding the direct relationship between world model accuracy and performance, we conducted an additional experiment using a **scripted** world model. This model generates perfect predictions with $1-\epsilon$ probability and random predictions with $\epsilon$ probability. The table below presents the episode rewards after 50 episodes, showing how the performance of DICP-AD in the Darkroom environment varies with the accuracy of the world model. Importantly, our approach is inherently robust, as it is **lower-bounded** with the "Without Planning" case by avoiding relying on the world model when it becomes unreliable. We designed this experiment to freely manipulate the accuracy of the world model, as the accuracy evolves over time steps in the main experiment, making it difficult to establish a direct relationship between the accuracy and the performance. We will include a related discussion in the revised version of our paper.
>
> | $\epsilon$ of Script World Model | Episode Rewards |
> |-------------|----------|
> | 0.00          |   15.925    |
> | 0.05          |   12.175    |
> | 0.10         |   8.350    |
> | 0.15         |    6.825  |
> | 0.20         |    6.825  |
> | 0.25         |    6.225  |
> | 0.30     |    4.825   |
> | Without Planning |    14.825    |
>
>
> > Q. Are there any other limitations?
>
> Aside from potential inaccuracies in the world model, we believe our method has no notable limitations compared to previous works [1], as our method uses the same data collection process and training parameter size, with only negligible additional computation.
> ***
>
> ### Regarding Inefficiency
>
>
> > Q. If the world model is inaccurate and the mimicking source algorithm is efficient then would it make DICP inefficient?
>
> An inaccurate world model can indeed lead to suboptimal model-based planning, which may reduce the efficiency of DICP. However, as mentioned earlier, the efficiency of DICP is lower-bounded. Additionally, if the source algorithm employs a highly efficient update rule, it could diminish DICP’s relative advantage. That said, as long as RL algorithms rely on gradient descent—given its inherently gradual nature—we believe there will still be opportunities for DICP to provide meaningful improvements.
>
> > Q. How often do inefficiencies of source algorithms cause inefficient learning in [1]?
>
> [1] demonstrates that naive distillation of learning histories introduces inefficiencies and that skipping intermediate episodes in these histories can result in faster learning compared to the source algorithm. Furthermore, our results show that combining DICP with [1] still enhances learning performance under the same dataset and parameter size, indicating that [1] retains some inefficiencies. This supports our argument that the inefficiencies of source algorithms **generally** contribute to inefficient learning in naive distillation and [1].
>
> > Q. With enough data for in-context learning, would this problem persist?
>
> The scenario described by the reviewer, where sufficient offline data is available for test tasks, is valid but falls outside the scope of our research. If **enough** offline data is available, the significance of online sample efficiency diminishes. In such cases, other learning approaches may be more suitable than meta-RL methods designed to enhance online sample efficiency.
>
> In contrast, when only limited offline data is available for test tasks, it could provide the policy with a better starting point for online learning. However, the underlying issue persists beyond this stage, as the learning capability of the distilled algorithm remains unchanged. Consequently, the problem continues to affect subsequent online interactions.
>
> > Q. What may be other alternatives to a model-based planning approach to solve this problem and why would model-based planning be a better solution to this problem?
>
> An alternative approach to addressing the inefficiency caused by the gradual updates of source RL algorithms is to skip intermediate episodes and use only every $n$-th episode in learning histories, as explored in [1]. This technique enables $n$-times faster policy updates than the source algorithm. However, such approaches require careful tuning of the skipping frequency based on the specific algorithm and its hyperparameters. In contrast, model-based planning is largely independent of the hyperparameters of the source algorithm, making it a more robust and straightforward solution.
> ***

---

> > ### Author Response · Authors · 2024-11-22
> > **Author Response to Reviewer XR4p**
> >
> > ### Computation Compared to Previous Work
> >
> > > Q. How does the computation of your approach compare to [1]?
> >
> > The FLOP count per action selection is summarized in the table below. Even the maximum value is negligible on modern GPUs, and the difference becomes even less significant when using architectures like IDT, which are specifically designed to handle longer sequences efficiently. Notably, this favorable trade-off aligns with the current trend of increasing inference-time computation to fully leverage the reasoning capabilities of Transformers, particularly through few-shot [2] and chain-of-thought prompting [3].
> >
> > | Method      | Darkroom | Dark Key-to-Door | Darkroom-Permuted | Meta-World |
> > |-------------|----------|------------------|-------------------|------------|
> > | AD          | 6M       | 20M              | 20M               | 709M       |
> > | DPT         | 6M       | 20M              | 20M               | 709M       |
> > | IDT         | 8M       | 8M               | 8M                | 3M         |
> > | DICP-AD     | 2G       | 18G              | 18G               | 8G         |
> > | DICP-DPT    | 2G       | 18G              | 18G               | 8G         |
> > | DICP-IDT    | 147M     | 147M             | 147M              | 15M        |
> > ***
> >
> > [1] Michael Laskin, Luyu Wang, Junhyuk Oh, Emilio Parisotto, Stephen Spencer, Richie Steigerwald, DJ Strouse, Steven Stenberg Hansen, Angelos Filos, Ethan A. Brooks, Maxime Gazeau, Himanshu Sahni, Satinder Singh, and Volodymyr Mnih. In-context reinforcement learning with algorithm distillation. In International Conference on Learning Representations, 2023.
> >
> > [2] Tom B. Brown, Benjamin Mann, Nick Ryder, Melanie Subbiah, Jared Kaplan, Prafulla Dhariwal, Arvind Neelakantan, Pranav Shyam, Girish Sastry, Amanda Askell, Sandhini Agarwal, Ariel Herbert-Voss, Gretchen Krueger, Tom Henighan, Rewon Child, Aditya Ramesh, Daniel M. Ziegler, Jeffrey Wu, Clemens Winter, Christopher Hesse, Mark Chen, Eric Sigler, Mateusz Litwin, Scott Gray, Benjamin Chess, Jack Clark, Christopher Berner, Sam McCandlish, Alec Radford, Ilya Sutskever, and Dario Amodei. Language models are few-shot learners. In Neural Information Processing Systems, 2020.
> >
> > [3] Jason Wei, Xuezhi Wang, Dale Schuurmans, Maarten Bosma, Ed Huai hsin Chi, F. Xia, Quoc Le, and Denny Zhou. Chain of thought prompting elicits reasoning in large language models. In Neural Information Processing Systems, 2022.

---

> > > ### Comment · Reviewer_XR4p · 2024-11-25
> > >
> > > Thank you for your responses.
> > >
> > > * To clarify:
> > >
> > >     * When ϵ = 0: The world model is perfectly accurate, and DICP-AD can potentially outperform the "Without Planning" case by leveraging the perfect information from the model.
> > >     * As ϵ increases: The world model becomes less reliable. While DICP-AD is designed to be robust and still provide some benefit, its performance might degrade. In some cases, relying solely on the "Without Planning" approach might be more efficient, especially if the world model's predictions are consistently misleading.
> > > The key takeaway is that the optimal strategy depends on the specific scenario and the reliability of the world model. DICP-AD offers a flexible approach that can adapt to varying levels of model accuracy, but it's important to consider the trade-offs between using the world model and relying on simpler strategies.
> > >
> > > The problem is that the accuracy of the world model is seldom known. The proposed approach performs better than the "without planning" case only when the world model is 100% accurate. Since DICP performs best in almost every experiment, inaccurate world models are not evaluated thoroughly. Overall, this looks like a good direction of research the authors have explored and needs deeper investigation and experimentation.
> > >
> > > * Naive distillation of learning histories is not a fair comparison (the world model can be inaccurate too for your approach in the same scenario). As mentioned earlier evaluations need further analysis. Introducing DICP is a unique idea and I value that. However, it is unclear to me that inefficiency in the source algorithm is the only reason for the boost in performance.
> > >
> > > As a result,  I believe my rating should stay at its current value.

---

> > > > ### Author Response · Authors · 2024-11-26
> > > > **Author Response to Reviewer XR4p**
> > > >
> > > > We sincerely appreciate the reviewer’s recognition of DICP as a unique and promising idea, and we would like to summarize our perspective in this discussion.
> > > >
> > > > - Inaccuracies in the world model are indeed a common limitation of most model-based planning methods. However, our framework stands out by being lower-bounded by the performance of model-free counterparts.
> > > > - Importantly, our method does not rely on having a perfect world model, which is particularly challenging in continuous dynamics settings like Meta-World. Despite this, our approach achieves state-of-the-art performance.
> > > > - Additionally, with its negligible computational overhead, our method remains both practical and effective across various scenarios.
> > > >
> > > > In response to the reviewer’s feedback, we will include further investigation in our revised manuscript and sincerely thank Reviewer XR4p for their insightful engagement and valuable suggestions.

---

### Official Review · Reviewer_jXdx · 2024-11-03

**Soundness:** 3
**Presentation:** 3
**Contribution:** 3
**Rating:** 6
**Confidence:** 3

**Summary:**

This paper proposes a model-based in-context reinforcement learning method called Distillation for In-Context Planning (DICP). With a dynamics model for planning, it provides the ability to deviate from the source algorithm's behavior. The authors show that DICP achieves better performance  on Darkroom and Meta-World benchmarks.

**Strengths:**

- The paper is clearly written, allowing readers to follow the main arguments.
- It provides a comprehensive experiments and ablations to demonstrate the effectiveness of DICP compared with the model-free counterparts.

**Weaknesses:**

I think the experimental results of this paper do not strongly support the main motivation. The authors claim that: “Model-free in-context reinforcement learning methods are trained to mimic the source algorithm, they also reproduce its suboptimal behaviors. Model-based planning offers a promising solution to this limitation by allowing the agents to simulate potential outcomes before taking action, providing an additional mechanism to deviate from the source algorithm’s behavior.” However, in the experiments section, DICP does not show significant performance advantages over its model-free counterparts. For example, as shown in Appendix B, the success rate of DICP-AD compared to AD only improves from 68% to 69%, and DICP-IDT compared to IDT only improves from 75% to 80%. Therefore, I believe model-based planning does not significantly enhance the policy beyond the source behavior.

**Questions:**

- Could the authors explain why improvements of DICP-AD over AD is not significant?
- Is it possible to evaluate on more challenging benchmarks like ML10?

---

> ### Author Response · Authors · 2024-11-19
> **Author Response to Reviewer jXdx**
>
> We sincerely appreciate reviewer jXdx’s constructive feedback. We hope this discussion will help bridge any gaps in understanding and further enhance the clarity of our work.
> ***
> ### Regarding Performance Gain
>
> **Steady Performance Gain**
>
> We would like to emphasize that our method achieves **steady** performance gains across a variety of models and environments, including DICP-AD, DICP-DPT, and DICP-IDT. While the magnitude of improvement varies, our models outperform their model-free counterparts in both discrete and continuous environments. Notably, DICP-IDT, our best-performing model, achieves the **state-of-the-art** performance on the well-established Meta-World benchmarks (Table 1).
>
> **Significantly Fewer Environment Interactions**
>
> The review primarily focuses on final success rate differences; however, we would like to highlight that **sample efficiency** is a critical consideration in RL. Our models learn faster than their model-free counterparts across both discrete (first row of Fig. 2) and continuous environments (last subfigure of Fig. 2), averaging across 50 tasks. Furthermore, Table 1 demonstrates that our approach achieves superior performance with significantly fewer environment interactions compared to extensive baselines. These results underscore the practical benefits of our method, particularly in reducing the cost of online data collection while maintaining strong final performance.
>
> **Agnostic to Sequence Model Choice**
>
> In our result, the performance improvement of DICP-AD over AD is smaller in ML1 compared to other settings. This difference is attributed to the **accuracy of the in-context learned world model**. Specifically, as shown below, the dynamics model in DICP-AD is relatively less accurate compared to DICP-IDT. This analysis indicates that the small vanilla Transformers used in AD may not be ideal for capturing long input sequences, whereas IDT incorporates design choices that better suit such tasks. Since the performance of model-based planning heavily depends on the accuracy of the learned world model, weaker sequence models inherently limit the gains achieved by our framework. It is important to note, however, that our method is **agnostic to the choice of sequence model**. As a result, our approach directly benefits from the use of advanced or scaled sequence models that can more accurately capture sequential dynamics.
>
> | | DICP-AD | DICP-IDT |
> |----------|----------|----------|
> | Test Dynamics Loss  | $8.9e^{-2}$  | $4.0e^{-2}$ |
> ***
>
> ### Evaluation on More Challenging Benchmarks
>
> In response to the reviewer's comment, we conduct additional experiments on the ML10 benchmark of Meta-World. The meta-test success rates below demonstrate that our approach outperforms the model-free counterpart and achieves state-of-the-art performance on this benchmark. Notably, this is achieved with significantly fewer environment interactions and without relying on expert demonstrations or task descriptions for test tasks. We will   incorporate the results into the revised version of our paper.
>
> | Method             | Success Rate       | Steps   |
> |--------------------|--------------------|---------|
> | PEARL             |                $13.0$ |   $350M$ |
> | MAML              |                $31.6$ |   $350M$ |
> | RL$^2$            |                $35.8$ |   $350M$ |
> | IDT               |                $36.7$ |   $500K$ |
> | DICP-IDT (Ours)   |        $\textbf{46.9}$ |   $500K$ |

---

> > ### Comment · Reviewer_jXdx · 2024-11-25
> >
> > Thanks for the authors' detailed explanation. I have no further questions and I'm willing to raise my score.

---

> > > ### Author Response · Authors · 2024-11-25
> > > **Author Response to Reviewer jXdx**
> > >
> > > We deeply appreciate the reviewer’s supportive feedback and decision to raise the score. We hope this exchange will lead to valuable contributions to the broader community.

---

### Author Response · Authors · 2024-11-25
**Author Response**

Dear Reviewers and AC,

We sincerely appreciate your invaluable feedback, which has been instrumental in improving our paper. In response, we have made the following updates:
- Added a computational analysis in Appendix A.
- Included an additional experiment on ML10 in Appendix B.
- Conducted an ablation study analyzing the impact of world model accuracy on final performance, detailed in Appendix C.

We hope these updates enhance the clarity and completeness of our work.

---

### Meta-Review · Area_Chair_qpdG · 2024-12-27

**Metareview:**

The authors present DICP, which builds upon decision transformers to enable in-context, model-based planning/RL.

The paper leverages model learning to accelerate and improve in-context adaptation to new tasks in RL. The method is general-purpose and the results show strong performance.

Reviewers highlighted several weaknesses. In particular, there were questions around whether all claims in the paper were justified by experimental evidence. There were also questions around the robustness of the learned model, although the authors added new results demonstrating this.

All reviewers recommend acceptance, and the authors have largely addressed the major weaknesses mentioned.

**Additional Comments On Reviewer Discussion:**

The most substantial point of discussion was on the robustness of learned models. While there was no conclusion about the robustness of models (and the impact of model error on performance), this is more of a "nice-to-have" than a critical part of the paper.

---

### Decision · Program_Chairs · 2025-01-22

Accept (Poster)